# Improved Management of Grassland to Promote Sustainable Use Based on Farm Size

Xin He [1,2,*], Jingru Wei [2], Suhua Gu [2], Luping Wang [3], Zechen Tian [4], Danqiong Chen [5] and Jiazhi Yuan [2]

1 Institute of Resources and Energy Research, Baotou Teachers' College, Inner Mongolia University of Science and Technology, 3 Science Road, Baotou 014030, China
2 School of Economics and Management, Baotou Teachers' College, Inner Mongolia University of Science and Technology, 3 Science Road, Baotou 014030, China
3 Department of Forest and Wildlife Ecology, University of Wisconsin-Madison, 1630 Linden Dr., Madison, WI 53706, USA
4 Faculty of Art and Science, University of Toronto-St. George Campus, 27 King's College Cir, Toronto, ON M5S 1A1, Canada
5 School of Economics and Management, Northeast Forestry University, 26 Hexing Road, Harbin 150006, China
* Correspondence: 66209@bttc.edu.cn; Tel.:+86-0-13347183629

**Abstract:** Grassland farms form the basis of grassland resource management in China. Farm sizes in China are generally small, which obviously increases the risk of grassland ecosystems. It is necessary to analyze the impact of farm size on grasslands from the perspective of livestock production in order to improve grassland management. This study combines field investigations and statistical analysis from 2004 to 2020, using a total of 126 farms from the Xilinguole League of Inner Mongolia in China as samples. These sample farms are divided into large farms and small farms. Different production scale and management behaviors are explored, along with their different impacts on grassland resources use. The results show that the expansion of farm size is constrained by the government management policies. Different behaviors are adopted by large and small farms in terms of finance, grassland circulation, and overgrazing management. The differentiation mechanisms of different farm size and the utilization of grassland resources are clarified in this study. This work suggests that managers promote sustainable use based on farm size and build appropriate policies to avoid future risks. The results of this study can provide a framework for solving similar problems.

**Keywords:** differentiation mechanism; grassland resources; farm size; adaptive management





## 1. Introduction

Natural grassland serves as an important animal husbandry base and ecological barrier against erosion in China. In the late 1960s, most areas of grassland in northern China were degraded, with frequent natural disasters. Natural disasters and the unsustainable use of grassland resources are some of the most pressing environmental problems in China. The unsustainable use of grassland resources, such as overgrazing and disorderly reclamation, has been found to be the main cause of grassland degradation and soil degradation [1,2]. Grassland mismanagement is commonly recognized as an important underlying driver of the unsustainable use of grassland resources [3–5]. In 1982, China implemented "Rural Land Contract System" in grassland pastoral areas, which requires that grasslands be allocated to families according to their population [6]. From then on, family farms on the grassland formed the basis of grazing land management and livestock production [7]. In the meantime, the government has implemented a range of policies, such as the "Measures for the Balanced Management of Grass and Livestock", rest period on a specific land areas, seasonal stocking, restricting grazing for a year to allow for restoration, and encouraging regional grassland rotation [6,8,9].

With the transition from traditional animal husbandry to commercialization, the deterioration of grassland ecosystems has become increasingly prominent. In China, research on

grassland management of land–forage–animal relationships is divided into three categories: research related to soil and grass (A interface); research related to grassland and animals (B interface), and research related to grassland livestock production (C interface). At C interface, human management of grassland is of particular value [10,11]. Both ecologists and economists have pointed out that grassland management in northern China is unsustainable [12–14]. Some policies, such as forbidding grazing, as well as the rest period of grasslands, have directly affected livestock production [15] and the incomes of herders [16]. Since 1982, families living on the grassland have obtained livestock and grassland according to their family population, and the herd size of farms is much smaller than in previous years. For herders, herd size is very important because the more animals they sell the greater their income is. According to the policy, small herd size corresponds to small grassland area, which is not conducive to grazing management. The problem of the contradiction between the grassland area and number of livestock in farms has arisen, and the difference between larger farms and smaller farms has also emerged. Some researchers have observed that there are differences not only in production, but also in the way grassland is used between large and small farms [17]. Some large farms are formed due to large family populations, but some large farms are formed by renting grassland from other families. Herders can rent grassland to expand production and raise more animals, but rented grasslands tend to be overgrazed, known as "rental grazing hazard" [18]. Based on the analysis of grassland management and livestock distribution at a village scale, research has shown that it is impossible to find appropriate measures for grassland degradation without analyzing overgrazing in terms of specific spatial–temporal scales such as farm size. Changes in farm size caused by changes in property rights have played an important role in this degradation, it is defined as "distributed overgrazing", which means that the fragmentation utilization of grassland by herders has led to the overgrazing and degradation of part of grasslands [19]. In management of grazing lands, larger farms have stronger financing ability, making them more flexible when constrained by grassland management policies [20]. In the future, livestock production will be increasingly affected by competition for natural resources [21]. China's strategy is to develop modern animal husbandry [22]. In the Asian Dryland Belt, approaches to meet this demand have focused on grazing intensification and large-scale livestock production [23]. Livestock type and scale management are more crucial than grazing intensity for grassland conservation [24,25]. In the grasslands of northern China, farms are relatively small in terms of productivity and management of grazing land, while a lack of production funds is considered to be their "largest difficulty in production" [26]. Paddock management is common, herders often feed animals the maximum quantity allowed by the policy to obtain the greatest benefits in their farms. Maintaining a balance between available grassland and livestock stocking rates is the premise of the sustainable use of natural grassland and efficient grazing livestock production [27]. The policy of reducing the number of livestock and strengthening grassland management in pastoral areas of China has achieved certain goals. Over the past 20 years, great advances have been made in terms of both the economic development and ecological protection of pastoral areas in Inner Mongolia [28]. A thorough discussion of the current practical problems, such as farm size, will help the government refine policies, solve specific conflicts, and avoid potential risks.

In previous works on the management of grassland, the impact of farm size has been emphasized; however, although it has often been treated as an important parameter or classification method, there is still a lack of specialized analysis on management of grasslands related to farm size. Small grassland farms are considered to have management risks to the grassland ecosystem, but are large farms ecologically safe? How does the farm size affect herders' management? Are herders' management of grazing land sustainable in a grassland farm? This study carries out a realistic analysis of the limitations and differences of farm size in northern China, and analyze the differences and impacts of management related to farm size. The objective was to compare economic behaviors (financing, grassland circulation, and overgrazing management) and utilization of production factors (grasslands,

livestock capital, and husbandry labor) between large farms with more than 500 animal unit and small farms less than 500 animal unit. In this study, we can identify some grazing management ways and risks related to farm size, so as to provide research evidence for grassland ecologists and economists, and provide valuable management strategies for grassland managers to adjust policies.

## 2. Materials and Methods

### 2.1. Research Area

The present study analyzed a selected research area within Baiyinxile Ranch in Xilinguole League, Inner Mongolia (hereafter referred to as the "research area"). This research area lies on the edge of the low hilly area in the southeast of the Xilinguole grassland, with typical grassland covering an area of 373,000 hm$^2$. The local economy is dominated by animal husbandry, which follows the grazing production system, and mutton and dairy are the main products. A contracting period of 30 years was established for grassland and livestock, and families were allocated the right to manage animal husbandry in 2003. From 2015 to 2017, the grassland right verification policy was carried out to stabilize the grassland contracting relationship and encourage grassland circulation. "Measures for the Balanced Management of Grass and Livestock" were formulated according to the "Grassland Law". This policy requires that livestock stocking rates should match the grassland area of a farm. This has been implemented nationwide since 2005, and the main management objects are family farms and other operators in the grassland. In 2020, this policy was suspended temporarily. The provincial governments have formulated local management policies in accordance with the basic principles of previous management policy. "Measures for the Balanced Management of Grass and Livestock" serves as the most influential policy in the field of ecology and economy in this research area.

According to statistical analysis on the research area, the average livestock number in farms is 413.69, and the average labor force is 6.46, including 4.33 family members and 1.73 long-term employees. In the research area, herders usually think that farms with more than 500 livestock are "large" farms, because their most obvious characteristics are large grassland area and strong capital management and development ability. In contrast, "small" farms with less than 500 livestock are prone to financial difficulties. It can be seen from the grassland allocation data in 2003 and 2005 that families with large population obtained more livestock and larger allocated grassland areas, while families with small population obtained fewer livestock and fewer allocated grassland areas. These smaller farms often cooperate with relatives to increase the grassland area for scale benefit. According to local living standards, it is difficult for small farms to attain the necessary extra capital to buy more ewes for expanding production. After the conformation of farm patterns in 2003, with the implementation of grassland circulation and the "Measures for the Balanced Management of Grass and Livestock", obvious differences were observed between various farms. Some families rent more grasslands, raise more livestock, and have stronger management skills. Some families rent out their grasslands and leave pastoral areas. In the research area, large farms were found to tend to continuously expand to larger-scale production, while small farms tended to shrink or maintain production scale due to their limited operating abilities. This has caused small farms to become smaller and large farms to grow even larger, with clear separating boundaries and trends. Herders of large and small farms have markedly different resource utilization behaviors related to animal husbandry production and grassland resource management.

### 2.2. Data Sources and Indicator Specifications

The research data employed here combine field investigations and official data. Investigation data were obtained during September 2006, August–September 2008, August–November 2009, August–October 2015, August–November 2018, and August–November 2020, and included a basic description of grassland area in farms, livestock production factors, grassland circulation, and financing. A total of 126 farms were surveyed. Official data

included financial data related to livestock production and sales, the quantity of livestock, and grazing lands management from 2004 to 2020. In the present study, statistics related to 126 farms for 13 years (from 2004 to 2016) were used to form 1638 basic panel data points. During this period, the production scale and mode of the farms were representative.

Herd size refers to the number of livestock at a farm. Under the management policy, herd size generally corresponds to the grassland areas and the farm size. According to the data of livestock and output values in the research area from 2004 to 2020, the proportion of sheep and goats to the total number of livestock was 88.26–95.32%, and the proportion of mutton production value to the total grazing output value was 69.45–93.46%. Therefore, in the present study, one head of sheep was used as the animal unit according "Measures for the Balanced Management of Grass and Livestock" policy, and other livestock such as goats, cows, horses, donkeys, mules, and camels were calculated as 1–6 sheep. In the present study, the calculation of weight ($Q_h KG$) is as follows:

$$Q_h KG = (E_h PCS + S_h PCS) \times \frac{Q_g KG}{Q_g PCS} \tag{1}$$

where $Q_h KG$ is calculated in kg, $E_h PCS$ is the number of livestock slaughtered and eaten by herders, $S_h PCS$ is the number of livestock for sale, $Q_g KG$ is the annual output as calculated by weight (kg) in the research area, and $Q_g PCS$ is the number of livestock in the research area.

Overgrazing refers to a kind of stocking method in which the number of animal unit in the farm exceeds the policy requirements. According to the "Measures for the Balanced Management of Grass and Livestock", if the number of animal unit exceeds the approved load, it can be balanced in several ways: establishing a forage base, purchasing forage, rearing livestock in barns, selling more animals, or renting more grasslands. In the research area, there are two cases of overgrazing management, one is to adjust the number according to the policy to meet the requirements, the other is to manage at the cost of breaking the law.

*2.3. Sample Farm Production Function*

The main economic activities in the research areas were related to producing livestock products by investing in production factors, such as grassland, livestock capital (e.g., ewes), and husbandry labor. Based on the panel data of 126 farms from 2004 to 2016, the present study constructed a Cobb–Douglas production function model of sample farms using the following three input factors: grasslands (A), livestock capital (K), and husbandry labor (L), as the explanatory variables. A linear transformation was carried out to obtain the logarithmic linear regression model of farm production function. Assuming that a significant difference exists among farms, a fixed impact model was used to distinguish the difference between farms with virtual variables. EVIEWS 7.2 was used to centralize the farm data. A model (2) for the new variables was established,

$$Q_{it}^* = \beta_A A_{it}^* + \beta_K K_{it}^* + \beta_L L_{it}^* + \varepsilon_{it}^* \tag{2}$$

where i is the i$^{th}$ farm; *t* is the year; $Q_{it}^*$ is the output (kg) of the i$^{th}$ farm in year t; $A_{it}^*$ is the number of ha (hm$^2$) of the i$^{th}$ farms in year t; $K_{it}^*$ is livestock capital calculated from ewe data (each) of the i$^{th}$ farm in year *t*; $L_{it}^*$ is husbandry labor calculated in days (D) of the i$^{th}$ farm in year t; $\beta_A$, $\beta_K$, and $\beta_L$ are the elasticity coefficients of $A_{it}^*$, $K_{it}^*$, and $L_{it}^*$, respectively; and $\varepsilon_{it}^*$ is the random error term.

*2.4. Analysis Methodology*

The multi-level subdivision of the farm size is meticulous, but from the perspective of eco-economic system management and the trends in grassland husbandry scale development, the division into two groups of large and small farms is obviously more realistic and efficient. The present study paid more attention to the direction of the scale-based development of farms. Considering the status of herd size change and local living standards in the

research area, 126 farms were divided into large and small groups. In the research area, the number of animal unit in each farm from 2004 to 2016 was calculated. In ascending order, farms were divided into either a small farm group with less than 500 animal unit, or a large group with more than 500 animal unit. This was simply a relative division method used to analyze trends and differences with regard to the local situation, and does not represent an absolute demarcation. The small groups included 81 (64.29%) farms, while the large groups included 45 farms (35.71%). According to the field investigation data, this research specially clarifies the differences between large and small farms with regard to financing, grassland circulation, and overgrazing management. Combining statistical analysis, a Cobb–Douglas production function model of sample farms was established, and a regression analysis of the model was carried out. The differences between the large and small farm groups, with respect to the inputs of grassland, livestock capital, and animal husbandry labor, were compared.

## 3. Results

### 3.1. Different Behaviors of Farms in Financing, Grassland Circulation, and Overgrazing Management

#### 3.1.1. Expanding Production Scale by Financing

The field investigations from 2006 to 2015 showed that it was very common for grassland herders to borrow money for production, this financing behavior declined in 2018 and 2020. Statistics shows that the average purchase price of mutton has been rising every year since 1997. The questionnaires in 2016 and 2020 showed that herders have been optimistic about the price of mutton and other livestock, and more than 90% of them hope to raise more sheep. Moreover, 90% herders said that, if they had enough money, they were willing to buy more ewes and forage, as well as rent more grasslands to expand production. This expectation remains the main consideration in interviews in 2018 and 2020. Apart from the income from livestock production, the main financing channels of farms in the research area are bank loans, cooperative operations, loans from relatives or individuals, and private credit loans, among which the main sources of funds are loans from relatives and private credit loans. In pastoral areas characterized by financing, livestock is regarded as an important credit collateral that gives farms with large herd size an obvious advantage over farms with small herd size when seeking money.

The behavior of herders to borrow money for production started from 2003 in the research area. From 2007 to 2009, persistent drought, pests, and rodents affected more than 90% of farms. Since the grassland right verification policy was established in 2015, almost every farm operates on loans. The hay price in the research area was CNY 0.20/kg in 2004, CNY 1.20 in 2010, CNY 5.50 in 2016, CNY 7.30 in 2018, and continue to rise in 2020, meaning that the average price increased over 30 times over 16 years, which has greatly increased both the cost of livestock production and the demand for cash flow in grazing management of animal husbandry. If the farm unable to raise funds to buy forage for reproduction cycles on time during stocking season, the farm will face the risk of shrinking in terms of production scale and losing income. Large farms can borrow urgently needed funds using their livestock as collateral, and therefore have stronger abilities to resist risks. However, small farms operate under the pressure of loan interest every year and are vulnerable to the temptation of using illegal usury. In terms of the source of loan funds, informal credit amounts (generally referred to as private credit) accounted for 67% of the cash value of all loans in 2006 in the research area. In 2009, informal credit accounted for 87% of all loans (with monthly interest rates of generally 2–3%), loans from relatives (low interest or interest-free) accounted for 9%, and bank loans accounted less than 4%. In 2015, informal credit accounted for 89% of loans, loans from relatives accounted for 4%, and bank loans accounted for 7%. In 2018, there was a significant decrease in informal credit amounts and an increase in bank loans. There are differences between small farms and large farms in terms of loan amounts, main loan channels, and the main use of loan funds (Table 1). In addition, in terms of loan period and repayment ability, small farms typically have a long

loan period and often fail to reimburse the loan in the same year, while large farms have more flexible, relatively active, and rational economic behavior.

**Table 1.** Manners in which farms financed production for small and large farm groups.

| Manners in Which Farms Financed Production | Year | Small Farm Group | Large Farm Group |
|---|---|---|---|
| Annual average amount of borrowed funds (thousand CNY) | 2006 2009 2015 | 9.32 22.60 31.20 | 34.30 64.60 62.90 |
| Main loan channels | | Loans from relatives | Private credit loans |
| Main use of loan funds | | Purchase forage, pay for grassland rent and living needs | Rent more grasslands, pay for grassland rent, and purchase forage |

### 3.1.2. Different Grassland Circulation Behaviors

In 2003, the Huanghuashute Branch of the research area provided grasslands for 291 farms. In October 2009, 168 farms were still engaged in animal husbandry and 123 farms (42% of all farms) had left for non-pastoral employment. In September 2015, only 142 farms were engaged in animal husbandry (when compared with 2003, the number decreased by 51%). Field investigations found that those members of farms who had left the pastoral area were mostly from small farms and that they had usually moved out after renting out their grasslands, especially after disasters in 2007–2009 and the grassland right verification policy in 2015. As of October 2020, 21 of the 126 families surveyed in the present paper have left the pastoral area, including 20 from small farms and 1 from a large farm, which provides further evidence that small farms face the risk of being squeezed out of the pastoral area. Some farms hire workers for the grazing and lamb midwifery periods while their non-labor family members live in cities or towns. As found in the investigation, grassland circulation has always been an important part of animal husbandry production because it can not only provide the necessary forage, but also increase herd size according to the "Balanced Management of Grassland and Livestock", thus making it legal to expand herd size. Most farms did not have enough forage to feed animals, so renting more grassland or buying more forage were common methods for supplementing forage. Small farms try their best to rent grasslands to keep their existing herd size in line with the requirements of the "Measures for the Balanced Management of Grass and Livestock". The essence of this rental behavior is to maintain livestock production and survive economically, while large farms rent more grasslands and increase herd size in the pursuit of profit. After the grassland right verification policy in 2015, grassland circulation became relatively active again, with some small farms renting out their grasslands and leaving grassland areas, and others renting grasslands to expand production scale.

In 2009, 82.5% of all farms rented grasslands from other farms. This fell to 74.6% by 2015. The cost of renting is increasing; the main use of leased grassland and the main forms of lease transactions in small and large farm groups are different (Table 2).

In the field investigations of 2009, 2015, and 2020, the average rental grassland areas of large farms were larger than those of small farms. After investigating the production operation of large farms, it was found that large farms had stronger financing ability. With more animals as collateral, large farms can obtain more money, so the total rental areas and average rental areas are greater than those of the small farms. The rental areas in 2015 and 2020 were larger for large farms than those in 2009. These data were not much different for small farms (Figure 1a). In the field investigations of 2018, a large farm rented out its grassland and left the research area due to a family affair, and 18 small farms left the research area, mainly due to economic difficulties in production (in the 2020 investigations, this figure was 20).

**Table 2.** Farm leasing for small and large farm groups.

| Farm Leasing | Year | Small Farm Group | Large Farm Group |
|---|---|---|---|
| Number of farms with leasing behavior | 2009 | 60 | 44 |
| | 2015 | 49 | 45 |
| Average rental area (hm$^2$) | 2009 | 153.9 | 345.3 |
| | 2015 | 170.4 | 411.5 |
| Price of leased grasslands (CNY/hm$^2$) | 2009 | 105–150 | 105–150 |
| | 2015 | 230–280 | 230–280 |
| Main use of leased grasslands | | Grazing | Grazing, mowing, and reserve for the balance policy |
| Main forms of lease transaction | | Cooperative operation | Cash payment |

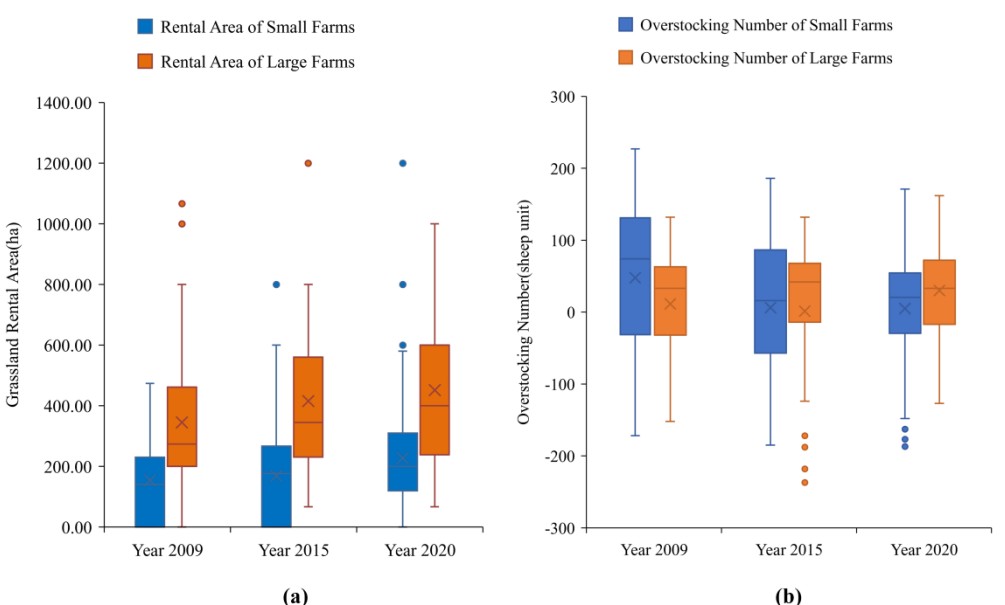

**Figure 1.** Box plots of grassland rental areas of large and small farms in 2009, 2015 and 2020 (**a**), and overgrazing numbers for large and small farms in 2009, 2015 and 2020 (**b**).

### 3.1.3. Overgrazing Management in Farms

Distributed overgrazing has occurred in the ecological landscape of grasslands due to fencing management and the overgrazing of some grasslands. This situation was further aggravated by the differential trends of large and small farms. Mutton prices continue to rise and herders have used grassland resources with great enthusiasm and efficiency for higher profits. Farms that are too small for rotational grazing are inevitably forced to overgraze inside and outside of fenced areas. The "Measures for the Balanced Management of Grass and Livestock" policy alleviates the grazing pressure on grassland, but places grazing management pressure on small farms. Small farms tend to stock at a high grazing pressure as a management strategy. Large farms tend to overgraze rented grasslands. Herders struggle to conceptualize the scale of rented grasslands and the efficiency of rotational grazing, and are likely to overuse parts of the grasslands. The differences between small and large farms may lead to potential grassland ecological risks.

As livestock prices rise, herders tend to raise more animals. Overgrazing can be adjusted by selling animals during the animal number checking period. This method of adjusting overgrazing by selling animals is used in both large and small farms (Figure 1b). Combined with the survey, we found that, in small farms, overgrazing and grazing below the prescribed stocking rate coexist, while the herd sizes in large farms are rarely lower than the prescribed stocking rate. Small farms mainly adjust overgrazing by selling animals, while large farms adjust overgrazing by renting more grasslands or purchasing more forage,

thanks to their stronger financial position. A relationship exists between overgrazing and the financial ability of farms. Under the grazing management pressure of maintaining their livestock, small farms are prone to continuous stocking or illegal overgrazing (i.e., exceeding the prescribed stocking capacity but not adopting the overgrazing balance method allowed by the policy), while overgrazing in large farms is often legitimate in terms of policy.

### 3.2. Differential Use of Production Factors

Ordinary least squares regression was carried out for all farms and for the two farm groups analyzed here separately; $\beta_A$, $\beta_K$, and $\beta_L$ for all farms were 0.7321, 0.3819, and 0.5762, respectively. This result shows that the output elasticity of livestock capital is smaller, while the output elasticity of grassland and husbandry labor is larger. Order $\sum \beta_i = \beta_A + \beta_k + \beta_L$, $\sum \beta_i = 1.6902 > 1$ (Table 3) shows that the rewards of production scale increase, i.e., the output develops at an increasing rate, indicating that it is beneficial for farms to expand production scale.

**Table 3.** Results of pool estimation * for small, large, and all farm groups.

| Estimation Output for: | Total Pool (Balanced) Observations | Coefficient $\beta_A$ (lnA) | Coefficient $\beta_K$ (lnK) | Coefficient $\beta_L$ (lnL) | $\sum \beta_i$ |
|---|---|---|---|---|---|
| All farms | 1638 | 0.7321 | 0.3819 | 0.5762 | 1.6902 |
| Small farm group | 1053 | 0.8329 | 0.3290 | 0.4192 | 1.5811 |
| Large farm group | 585 | 0.6403 | 0.4268 | 0.6846 | 1.7517 |

* Significance level is $p < 0.05$.

An analysis of variance (ANOVA) of $A_{it}^*$, $K_{it}^*$, and $L_{it}^*$ for the two farm groups was conducted. F values were greater than F crit values, and P values were greater than 0.01 and less than 0.05, indicating significant differences. The results of ordinary least squares (OLS) regression for the two groups separately showed that differences existed between them and were mainly manifested in the following aspects. First, differences existed in rewards of scale. In the large farm group, $\sum \beta_i$ was larger than in the small group (Table 3). These two groups both conducted scale expansion as their production strategy; however, because the rate of increase in output was larger in the large group than in the small group, the large group had a comparative advantage. From the perspective of economic profit, it is better to increase production in large farms at the same cost. Second, differences existed in factor elasticity. The elasticity of production factors in the large group was in the following order: husbandry labor > grassland > livestock capital. In the small group, the order was: grassland > husbandry labor > livestock capital.

Under the non-overgrazing management mode, the output elasticity of small groups is greater and, accordingly, grassland utilization is more intense. Herders of large farms have the willingness and ability to rent more grasslands and continuously expand their production scale. This development model has increased the management power of large farms, and has scale advantage, which is conducive to the implementation of seasonal stocking, stocking cycle, rotational grazing, and other eco-friendly grazing land management.

In terms of the husbandry labor force, the average long-term labor in large farm groups was 2.61, and that in small groups was 2.09. The average number of animals per capita in large groups was 182.63, and 92.06 in small groups. During the busy season, large farms often employ short-term workers for handling lambs, captivity feeding, weeding, sales, and other duties. Small farms sometimes need to hire short-term workers in the busy season, but in the off-season, there is often a labor surplus. Large farms use labor resources more efficiently, while small farms cannot make efficient use of the available labor.

According to the data of Inner Mongolia, although the annual income of farms has increased, the number of herders leaving the grasslands has also increased (Figure 2). Since 2003, more than half of the farms in the research area have left the grasslands for

non-pastoral employment. With the large number of herders leaving and the willingness of their descendants to follow, there is a shortage of husbandry labor in the research area.

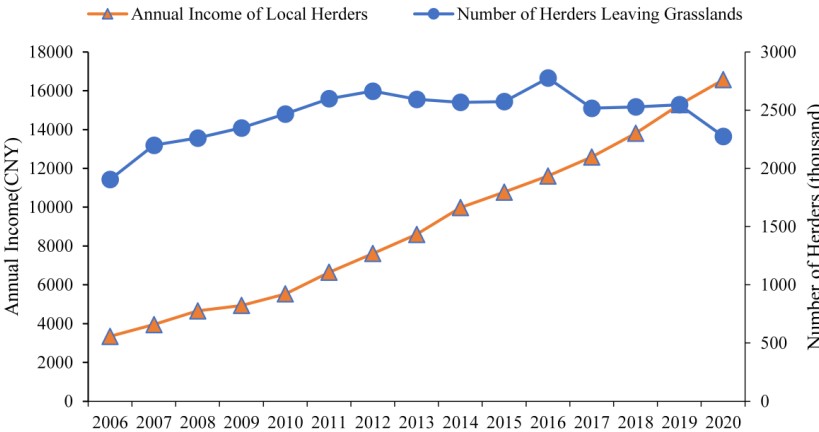

**Figure 2.** Number of herders leaving grasslands and annual income of herders in Inner Mongolia from 2006 to 2020.

## 4. Discussion

### 4.1. Differential Mechanisms of Grassland Resource Use

Based on the division of herd size for small and large farms, a typical internal mechanism has been found in the present paper, that is, the farm size affects the grassland by affecting the managements of herders. Farms of different sizes obviously have different management behaviors, which obviously increases the risk of grassland ecosystems, specifically, the differentiation mechanism of the utilization of grassland resources in farms. Different herd sizes affect the financing ability of farms. There are obvious differences between small and large farms because of differences in financing operations, the input of production factors, grassland resource use plans, stocking density, and other behaviours of grazing lands management, all of which contribute to the impact on the grassland ecosystem. The inherent feature of this mechanism is that livestock is a special asset. The external driving condition of this mechanism is market-oriented production and consumption, while the restrictive factors are the "Measures for the Balanced Management of Grass and Livestock" policy and other ecological restraint policies. The linkage of this mechanism is from herd size to financial operation, and then the livestock production behavior of farms, all of which ultimately have an impact on the grassland ecosystem. This is depicted in Figure 3.

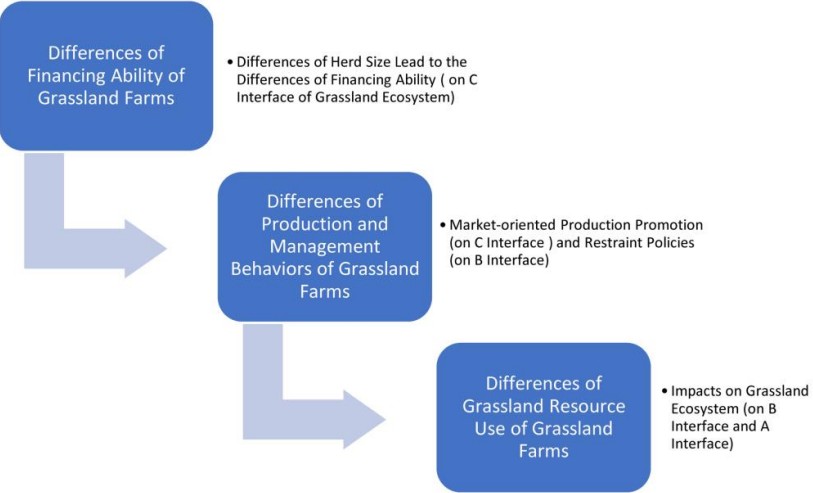

**Figure 3.** Differentiation mechanism for grassland resource use behavior of farms.

### 4.2. Overgrazing Management of farms

Under policy constraints, farms can adopt overgrazing or non-overgrazing methods. In recent years, many farms have adopted the overgrazing production mode, especially after 2007, when mutton prices increased rapidly. The overgrazing balancing measures allowed by the "Measures for the Balanced Management of Grass and Livestock" policy can be achieved in most large farms, while continuous stocking, grazing in forbidden areas, and illegal overgrazing often occur in small ones. Nevertheless, overgrazing in small farms is limited by economic ability, while the overgrazing risk of large farms cannot be ignored. The overgrazing characteristics of large farms are legality, large quantity, and seasonal variation, which increase the potential ecological risks. The measures to balance overgrazing, such as purchasing forage and selling livestock, are difficult to track and manage quantitatively. As the main management objects of the "Measures for the Balanced Management of Grass and Livestock" policy are farms rather than larger areas, overgrazing beyond the farm scope cannot be monitored. Especially since 2020, the provincial governments have formulated local management policies, and the grass transportation industry is developed, the overgrazing across provinces is also a potential risk. Therefore, it is still necessary to carry out unified management nationwide, and the quantitative method of overloading is necessary to investigate a wider space except farm scope. Additionally, it is easy to cause new ecological or financial risks in grassland areas due to rent-seeking or disorderly management. Regarding overgrazing, managers should not only prevent small farms from overgrazing grasslands, but also seek to prevent the ecological risks of large farms overgrazing at different spatial and temporal scales. Managers should also guard against rent-seeking, the distortion of statistics, and social disputes caused by the complexity of overgrazing management.

### 4.3. Grassland Circulation Bottleneck and Herders Leaving the Grasslands

At the beginning of the implementation of the grassland contracting policy, grassland circulation was active. However, after 2009, the amount of grassland circulation declined. The present study found that, aside from the limited total amount of grassland and the high rent costs limiting grassland circulation, the existence of transaction costs, such as grassland distance, renter preference, and lease negotiation, are also important factors. Tenants generally prefer to choose adjacent grasslands because grasslands further away are not convenient for grazing and mowing. In the early stages of the grassland contracting policy, some farms would rent out grasslands, while some left the pastoral area for non-pastoral employment or were employed by other large farms. In the later stages, some small farms decided to maintain their position on the grassland even though their production was inefficient, mainly as a result of their preference for herding work, or a lack of other job skills. As a result, the development mode of production expansion from renting grasslands has entered a bottleneck that has increased the utilization pressure on grasslands. It is difficult to improve the paddock production mode and the grazing distribution of grasslands. Even for large farms, the pastoral areas are still not large enough for grazing cycle over a wider range. If the production scale does not change, it will be difficult for pastoral areas to avoid the problems of distributed overgrazing and grassland fence dilemma.

### 4.4. Potential Risk Related to Farm Size

In grassland management, where farms form the basis of grassland resource management, special attention should be paid to the potential risks related to farm size. These risks are: farm size too small, farm size differences and farm size expansion processes. Farms with a small production scale rely heavily on grassland to bring benefits, falling into the dilemma of maintaining grazing management or livelihoods. The size differences between farms in pastoral areas have resulted in grassland segmentation, distributed overgrazing, and unbalanced resource allocation. Expanding the production scale through cooperative operation or creating new ranches has been carried out in pastoral areas in Inner Mongolia. There are potential risks in the process of farm size expansion, such as overgrazing in rented

grassland or the overuse of grassland resources across a wider range. The forage purchased during the expansion often comes from the adjacent grassland, which will have an impact on the grassland on a larger range. Meanwhile, the difficulty of grassland management caused by social factors related to the expansion of farms should not be ignored.

*4.5. Preparation for the Development of Larger Farms*

Many valuable observations related to market-oriented and grazing land management have been made based on the perspective of farms. In both the mutton and dairy industries, the movement towards larger and more modern farms is effective economically and ecologically; the movement has also received policy support. From an ecological viewpoint, the sizes of grassland farms in the research area are generally too small to realize the sustainable use of grassland resources. Along with realizing economic benefits of scale, large farms also see ecological benefits that come from the movement towards larger farm development. The problems that need to be solved are how to overcome the obstacles of grassland circulation, standardize overgrazing management, train modern husbandry labors, and promote the upgrading of animal products. Taking the differentiation mechanisms of the utilization of the grassland resources of farms into account, policies can be designed to encourage the modernization of large farms and to train modern labors from small farms. This will solve the problems of maintaining the livelihoods of herders and increasing their income in the context of urbanization in China.

**5. Conclusions**

In pastoral areas characterized by financing, livestock is regarded as an important credit collateral that gives large farms an obvious advantage over small farms. Data were collected from 2004 to 2020, using a total of 126 farms from Inner Mongolia to analyze differences in farm size and production, as well as the impact of internal mechanisms on grassland management. Farms were divided into large and small groups based on herd size, differences in finance, overgrazing management and production were analyzed. The field investigations from 2006 to 2015 showed that large farm herders had more flexible, relatively active, and rational financing behavior.

The questionnaires in 2016 and 2020 showed that herders in grassland in Inner Mongolia were optimistic about mutton price, and more than 90% of them hoped to raise more sheep. However, the increase of herd size is strictly restricted by the "Measures for the Balanced Management of Grass and Livestock" policy. Grassland circulation can contribute to the effective implementation of the grassland management policy, but the behaviors in large and small farms are obviously different. Large farms are often the lessee, while the small farms are the lessor.

In order to obtain greater economic benefits, farms have mostly adopted overgrazing production. The overgrazing balancing measures allowed by the "Measures for the Balanced Management of Grass and Livestock" policy can be achieved in most large farms, while continuous stocking, grazing in grazing forbidden areas, and illegal overgrazing often occur in small ones.

Ordinary least squares regression was carried out for all farms, and the results showed that rewards of production scale increased, indicating that it is beneficial to expand production scale. The results of OLS regression for the two groups separately showed that differences existed between them and were mainly manifested in rewards of scale and factor elasticity. The output elasticity of grassland in small farms is greater and, accordingly, the grazing rate is higher, and the grassland utilization is more intensive. Large farms use labor resources more efficiently, while small farms cannot make efficient use of the available labor.

Based on the division of small and large farms, a typical internal mechanism has been found in the present paper, specifically, the differentiation mechanism of the utilization of grassland resources in farms. Different herd sizes affect the financing ability of farms. There are obvious differences between small and large farms because of differences in financing

operations, the input of production factors, grassland resource use plans, stocking density, and other behaviours of grazing lands management. The linkage of this mechanism is from herd size to finance operation, and then to the livestock production behavior of farms, all of which ultimately have an impact on the grassland ecosystem.

We suggest that large farms undergo long-term planning to improve grazing lands management and avoid expansion risks, while small farms improve the profitability and utilization efficiency of labor resources. We also suggest that the government promote the expansion of production scale and pay more attention to the difference of stocking rate between large and small farms. The problems that need to be solved are overcoming the obstacles of grassland circulation, standardizing overgrazing management, training modern husbandry labors, and promoting the upgrading of livestock products. Managers should also guard against rent-seeking, the distortion of statistics, and social disputes caused by the complexity of overgrazing management.

This analysis prompts managers to pay special attention to the potential risks related to farm size: too small farm size, farm size differences, and farm size expansion processes. In particular, farm size expansion risk has not been valued in the past policies. We suggest that managers promote sustainable use based on farm size. We also suggest that ecologists and economists cooperate to analyze the production scale and its impact on grassland ecosystem. This work will contribute to the innovative management of grassland resources in similar regions.

**Author Contributions:** Conceptualization, X.H.; methodology, X.H. and J.W.; software, X.H. and J.W.; validation, X.H., J.W. and S.G.; formal analysis, X.H. and J.W.; investigation, X.H., J.W. and Z.T.; resources, X.H.; data curation, X.H. and L.W.; writing—original draft preparation, X.H., Z.T. and D.C.; writing—review and editing, X.H., J.Y. and L.W.; visualization, X.H.; supervision, X.H.; project administration, X.H.; funding acquisition, X.H. All authors have read and agreed to the published version of the manuscript.

**Funding:** This research was funded by the Institute of Resources and Energy Research at Baotou Teachers' College, Inner Mongolia University of Science and Technology (Grant No. BTTCRCQD2018-16 and Grant No. BSYHY202202) and the Inner Mongolia Philosophy and Social Sciences Foundation (Grant No. 2018NDB089).

**Institutional Review Board Statement:** Not applicable.

**Data Availability Statement:** Not applicable.

**Acknowledgments:** The authors would like to thank the authorities, residents and temporary residents of the research area who participated in our research. We would also like to thank Jianming Niu who did much of the basic work for this research project. We would especially like to express our gratitude to reviewers for their valuable suggestions.

**Conflicts of Interest:** The authors declare no conflict of interest.

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
