# Peer review of "Improved Management of Grassland to Promote Sustainable Use Based on Farm Size"

_agriculture, doi:10.3390/agriculture13020350_

Round 1
Reviewer 1 Report (Previous Reviewer 2)
Please see my attached comments & suggestions.

Author Response
Thank you for your suggestions. Please see the attachment.

Reviewer 2 Report (Previous Reviewer 4)
The author has made very careful revisions and all the problems pointed out have been well refined
Author Response
Thank you for your comments and suggestions.
Round 2
Reviewer 1 Report (Previous Reviewer 2)
Point 1: Line 31: farms formed the basis of grassland management and livestock production. It seems to me that farms were the basis of grassland management and livestock production many years before “The farm contract responsibility system” was introduced. Please explain. What is a farm? Was the grassland common land prior to this? How was the grassland managed before this system was introduced? Who was responsible for grassland management before & after?
Point 2: Line 33: Provide more information about grassland management practices before & after this policy was introduced. Define long-term grazing. This term can mean for years or for a much shorter period. Line 34: restricting grazing for a year, is the meaning forbidding grazing for a year?
regional grassland rotation. A region is understood to be a much larger area than an individual farm. The implication that there is a regional rotation seems unlikely, unless the land is common, in which case all farmers’ livestock would be moved as a large herd between regions. The implication here is that traditional animal husbandry does not lead to grassland degradation but commercialization does. Please provide a description of each system of grassland management & why they differ in effect on the grassland degradation.
Point 3: Line 38: research fields related to soil and grass (A interface); research fields related to grassland and animals (B interface); and research fields related to grassland livestock production (C interface). I suggest the words fields should be deleted because it has been introduced just previously in the sentence & does not need to be repeated. Also, please understand that the word field can also be an area of farm land. I do not understand the difference between interface B & C, both involve grassland & animals.
Point 4: Line 41: I suggest that it is not the grassland management in northern China that is unsustainable it is the balance between grassland production & animal kind & numbers. There are a variety of methods of managing grassland, but the primary factor leading to grassland degradation is the inherent carrying capacity of the grassland. I’m sure you are aware of the complexity of the relationship between grazing animals & grassland. At this point in your introduction I expect you to cite experiments that have been conducted to determine carrying capacity, & what factors affect carrying capacity. I expect to see some relevant results. Once a carrying capacity has been established then policies may be put in place to improve & reclaim your grasslands. I assume that the contradiction refers to striking an appropriate balance between area of grassland available & its productivity, & number of livestock being fed from the grassland. However, you write that herd size is smaller which implies that grassland is degraded or measures to halt grassland degradation are being successful.
Point 5: Line 50: the behavior of grassland use between large and small farms. The English does not make sense. Use does not have a behavior. Do you mean the way grassland is used, that is, how the herdsman manages his/her resources?
I suggest that the size & quality of the resources available to the herder must be described in a manner that can be applied to any size of farm. Carrying capacity must be determined & used to define appropriated numbers/kinds of livestock that can be sustained on the different areas of grassland & I think will be no different on large or small farms. Now, carrying capacity may differ drastically between farms of different herders, in other words, good managers may be able to manage grassland sustainably at a higher carrying capacity. It is my opinion that you need objective research carried out on your grasslands. If you do not have such research for your area than this is an absolute need before you can make recommendations on policy. It is not inherently obvious that management of rented land must lead to degradation. I suggest it is much more complex. I think a good manager can manage rented grassland equally well as the family grassland. If rental grassland is becoming degraded it is because carrying capacity is being exceeded. I suggest that you need to look at what is behind farm size in producing more degradation of grassland. I assume pressure on small farmers to overgraze is much higher & this pressure leads to overgrazing. Overgrazing is also very complex & the effect of season, year, state of the grassland etc all must be considered.
Point 6: Line 59: “distributed overgrazing”, please define.
Point 7:Line 66: farms are relatively small in terms of productivity. Please be careful in how the word small is used. You have been using it as size of farm, I think that is area & number of livestock. Productivity is a rate, that is, for instance, animals produced per unit area. If small herders had more capital, it is not self-evident that they would be more productive, because their limiting factor may be production not productivity. If capital is restricting productivity I understand that, not only is that their grassland area small, but the productivity or their animals is low. But, I wonder how more capital would help since they have little grassland. If the solution is to rent grassland, you have already pointed out that renting increases grassland degradation.
Point 8: I am reading your most recent changes & am pleased that I understand many things much better. I also can now see that it is not necessary that I understand everything because to do that I could look at the references. For instance, “The farm contract responsibility system” I do not fully understand the system, but, for purposes of the Article I think I do not need to know. It is sufficient to know that this system resulted in a change & made the grasslands more sustainable.
Now I wish to ask you about the objective or objectives of your research. How your introduction is written does not provide a concise statement of objectives. Size of farm is central to your objective & I think you choose to divide farms into large & small. Obviously it is useful to classify farms according to size, but, according to line 83/84 you write that farm size has been found to have an impact on grassland management or ecosystem. Which do you mean, management or ecosystem? Management is the way herders treat their livestock & grassland & can be described. Ecosystem is much broader & more complex & the description is made up of multiple interacting factors, some related to herders, some to livestock, some to plants, some to soil, weather; all interacting & complex. I believe farm size per se has little impact on the ecosystem, except as it affects management. All grassland farms will present a risk to the ecosystem. All need to be managed using methods that emphasize the sustainability of the grassland. Line 88 is, in my mind, irrelevant since all human activity presents a risk to the environment. So, the objective? In a concise form. The objective is a key component of you article & must be precise, clear, concise & avoid ambiguities. I do not find a statement of objectives adequately expressed in lines 83 to101.
Point 9: I wish to have a well written statement of objective(s). This should be one or two sentences long. It must clearly state the basic or central reason explaining why you carried out your research. Please send this to me with nothing more.
Point 10: You place grazing management as central to your overall objective. However, in your specific objective you then include economic management behaviors & utilization of production factors as central. In both overall & specific objective you are comparing large & small farms.
Just because I sent you an example of my own does not necessarily mean I think you should have both an overall & a specific objective. Now I do not understand well what you mean by “economic management behaviors & utilization of production factors” but I think “grazing management” is only one factor in “economic management behaviors & utilization of production factors”. If that is the case then it would make more sense to me to put “economic management behaviors & utilization of production factors” as the overall objective & “grazing management” as the specific objective, but I wonder if you need to refer to grazing management except as one factor (see below).
Next, allow me to discuss the phrase “economic management behaviors”. When I search in Google for this phrase I obtain, first, “Behavioral economics” a mixture of economics & human behavior. I also find Behavioral Economics & Management. However, since I am not an expert in behavior, economics or management I suggest you use either economic behavior or economic management, not both.
Now, allow me to discuss “utilization of production factors”. This is clear to me. There are multiple factors that affect production, soil quality, soil management, kind of grassland, animal units & feed source, grassland management, grazing management etc. I have put grazing management as just one factor within your specific objective. This is why I do not understand how grazing management can be in your overall objective. Grazing management is not necessarily the most important factors influencing utilization of production at the level of farm or landscape.
I think economics & farm management are more important in your research than grazing management. In these areas of research I am no help to you. I assume you base your research on the fact that there are differences in grazing management between large & small farms but you do not need to describe what these differences are. If my assumptions are correct then I think you may complete the paper with short mention of grazing management.
Author Response
Response to Prof. William Bryan’s
Comments in Emails
ID: agriculture-2055851
Point 1: Line 31: farms formed the basis of grassland management and livestock production. It seems to me that farms were the basis of grassland management and livestock production many years before “The farm contract responsibility system” was introduced. Please explain. What is a farm? Was the grassland common land prior to this? How was the grassland managed before this system was introduced? Who was responsible for grassland management before & after?
Response 1: “farms formed the basis of grassland management and livestock production” has been replaced by “family farms on the grassland formed the basis of grazing land management and livestock production”. For the terminology translation of some policies implemented by the government, we have specifically consulted the official literature, revised the "The farm contract responsibility system" to "rural land contract system", and made a brief explanation of the main contents of the policy in the new version. We added special explanations to the “Measures for the Balanced Management of Grass and Livestock” policy, pointing out that: This (policy) has been implemented nationwide since 2005, and the main management objects are family farms and other operators in the grassland (Before that, the main object of grassland management was villages). In 2020, this policy was suspended temporarily. The provincial governments have formulated local management policies in accordance with the basic principles of previous management policy.
Point 2: Line 33: Provide more information about grassland management practices before & after this policy was introduced. Define long-term grazing. This term can mean for years or for a much shorter period. Line 34: restricting grazing for a year, is the meaning forbidding grazing for a year?
regional grassland rotation. A region is understood to be a much larger area than an individual farm. The implication that there is a regional rotation seems unlikely, unless the land is common, in which case all farmers’ livestock would be moved as a large herd between regions. The implication here is that traditional animal husbandry does not lead to grassland degradation but commercialization does. Please provide a description of each system of grassland management & why they differ in effect on the grassland degradation.
Response 2: Some terms are improperly used, which makes it difficult to understand our manuscript. Many inappropriate expressions in this manuscript have been deleted, such as “forbidding long-term grazing”, “grassland resting”, “family size”, “farm size expansion”, “associated operation rights”, “livestock number (unit)”, “livestock scale”, “ecological pressure”, “economic pressure”, “overgraze grasslands”, “ increasing scale”, “zonal rotation”, “production measures”, “full use”, “the intensity of grassland use”, “grassland management methods”, “ from an economic management perspective”,“overtime grazing”, “grazing in prohibited areas”,“grassland use is more intense”, etc.
We have used some new terms and expressions, such as “grazing land management”, “management of grazing lands”, “family population”, “rest period”, “seasonal stocking”, “Land-forage-animal relationships”, “Paddock”, “the animal unit”, “stocking season”, “stocking rate”, “stocking cycle”, “rotational grazing”, “stocking density”, “continuous stocking”, “grazing in forbidden areas”, etc. We revised the sentences to make these expressions as accurate and understandable as possible.
When we said: “With the transition from traditional animal husbandry to commercialization, the deterioration of grassland ecosystems has become increasingly prominent”, we don't think that traditional animal husbandry does not lead to grassland degradation but commercialization does, we just think that commercialization is one of the reasons for accelerating grassland degradation.
Point 3: Line 38: research fields related to soil and grass (A interface); research fields related to grassland and animals (B interface); and research fields related to grassland livestock production (C interface). I suggest the words fields should be deleted because it has been introduced just previously in the sentence & does not need to be repeated. Also, please understand that the word field can also be an area of farm land. I do not understand the difference between interface B & C, both involve grassland & animals.
Response 3: We have revised the description of this paragraph as follows “In China, research on grassland management of Land-forage-animal relationships is divided into three categories:” and “fields” has be deleted. We revised the sentence to explain the difference between interface B & C “At C interface, human management of grassland is of particular value”. That is, at the C interface, human management is added.
Point 4: Line 41: I suggest that it is not the grassland management in northern China that is unsustainable it is the balance between grassland production & animal kind & numbers. There are a variety of methods of managing grassland, but the primary factor leading to grassland degradation is the inherent carrying capacity of the grassland. I’m sure you are aware of the complexity of the relationship between grazing animals & grassland. At this point in your introduction I expect you to cite experiments that have been conducted to determine carrying capacity, & what factors affect carrying capacity. I expect to see some relevant results. Once a carrying capacity has been established then policies may be put in place to improve & reclaim your grasslands. I assume that the contradiction refers to striking an appropriate balance between area of grassland available & its productivity, & number of livestock being fed from the grassland. However, you write that herd size is smaller which implies that grassland is degraded or measures to halt grassland degradation are being successful.
Response 4: We agree with “the primary factor leading to grassland degradation is the inherent carrying capacity of the grassland” , so we analyzed Overgrazing in farms. In the section of Data Sources and Indicator Specifications, we added the explanation of the word Overgrazing according to the explanations in some academic papers. Word Overgrazing used in our manuscript refers to a kind of stocking method in which the number of animal unit in the farm exceeds the policy (Measures for the Balanced Management of Grass and Livestock) requirements. We added “ According to the policy, small herd size corresponds to small grassland area, which is not conducive to grazing management. ”on Line 51-53 to explain the use of herd size. We also added “Under the management policy, herd size generally corresponds to the grassland areas and the farm size. ” on Line 154-155. Small herd size means small grassland areas, which is more difficult to manage rather than reduce the pressure on grassland. We have unified the statement of policy “Measures for the Balanced Management of Grass and Livestock”. In fact, this policy has been officially issued several times in China, and we simplified it into one statement in this manuscript.
Point 5: Line 50: the behavior of grassland use between large and small farms. The English does not make sense. Use does not have a behavior. Do you mean the way grassland is used, that is, how the herdsman manages his/her resources?
I suggest that the size & quality of the resources available to the herder must be described in a manner that can be applied to any size of farm. Carrying capacity must be determined & used to define appropriated numbers/kinds of livestock that can be sustained on the different areas of grassland & I think will be no different on large or small farms. Now, carrying capacity may differ drastically between farms of different herders, in other words, good managers may be able to manage grassland sustainably at a higher carrying capacity. It is my opinion that you need objective research carried out on your grasslands. If you do not have such research for your area than this is an absolute need before you can make recommendations on policy. It is not inherently obvious that management of rented land must lead to degradation. I suggest it is much more complex. I think a good manager can manage rented grassland equally well as the family grassland. If rental grassland is becoming degraded it is because carrying capacity is being exceeded. I suggest that you need to look at what is behind farm size in producing more degradation of grassland. I assume pressure on small farmers to overgraze is much higher & this pressure leads to overgrazing. Overgrazing is also very complex & the effect of season, year, state of the grassland etc all must be considered.
Response 5: We rewrote line 50 as “Some researchers have observed that there are differences not only in production, but also in the way grassland is used between large and small farms”. In order to be easily understand, we explained “In the field investigations, we found that farm size affects the grassland by affecting the managements of herders.” on Line 92-93. In the introduction section, we added sentences“Small grassland farms are considered to have management risks to the grassland ecosystem, but are large farms ecologically safe? Are farm size differences and farm expansion processes ecologically safe? How does the farm size affect herders' management? Are herders' management of grazing land sustainable? In the field investigations, we found that farm size affects the grassland by affecting the managements of herders. ” to strengthen our research purpose and lead to the importance of our research.
In discussion section, In order to explain the difference in the management of large farms and small farms, we added “The measures to balance overgrazing, such as purchasing forage and selling livestock, are difficult to track and manage quantitatively. As the main management objects of the “Measures for the Balanced Management of Grass and Livestock” policy are farms rather than larger areas, overgrazing beyond the farm scope cannot be monitored. Especially since 2020, the provincial governments have formulated local management policies, and the grass transportation industry is developed, the overgrazing across provinces is also a potential risk. Therefore, it is still necessary to carry out unified management nationwide, and the quantitative method of overloading is necessary to investigate a wider space except farm scope. “
As you said, good managers may be able to manage grassland sustainably at a higher carrying capacity. I am glad for you brought up the question. Previous studies in China believed that it is easier to implement sustainable management or higher carrying capacity in relatively large farms, but we find that risks were also accompanied. This is one of our research values. After years of observation and investigation, we found that there are not only strong management capabilities but also potential risks in large farms and farm expansion process (encouraged by the government), and we explained this in the conclusion section of this manuscript and put forward our views and suggestions.
Point 6: Line 59: “distributed overgrazing”, please define.
Response 6: We rewrote this sentence as “Changes in farm size caused by changes in property rights have played an important role in this degradation, it is defined as“distributed overgrazing”, which means that the fragmentation utilization of grassland by herders has led to the overgrazing and degradation of part of grasslands.”
Point 7:Line 66: farms are relatively small in terms of productivity. Please be careful in how the word small is used. You have been using it as size of farm, I think that is area & number of livestock. Productivity is a rate, that is, for instance, animals produced per unit area. If small herders had more capital, it is not self-evident that they would be more productive, because their limiting factor may be production not productivity. If capital is restricting productivity I understand that, not only is that their grassland area small, but the productivity or their animals is low. But, I wonder how more capital would help since they have little grassland. If the solution is to rent grassland, you have already pointed out that renting increases grassland degradation.
Response 7: In the research method section, we explained the size of farm: ”In ascending order, farms were divided into either a small farm group with less than 500 animal unit, or a large group with more than 500 animal unit. This was simply a relative division method used to analyze trends and differences with regard to the local situation, and does not represent an absolute demarcation. “ In our manuscript, a small farm refers to a farm where the number of animal unit is less than 500. When we visited the herders, they said that if they had enough money, they would buy more animals and rent more grassland, which is the most common way to apply their funds. This is their most realistic behavior in production.
Point 8: I am reading your most recent changes & am pleased that I understand many things much better. I also can now see that it is not necessary that I understand everything because to do that I could look at the references. For instance, “The farm contract responsibility system” I do not fully understand the system, but, for purposes of the Article I think I do not need to know. It is sufficient to know that this system resulted in a change & made the grasslands more sustainable.
Now I wish to ask you about the objective or objectives of your research. How your introduction is written does not provide a concise statement of objectives. Size of farm is central to your objective & I think you choose to divide farms into large & small. Obviously it is useful to classify farms according to size, but, according to line 83/84 you write that farm size has been found to have an impact on grassland management or ecosystem. Which do you mean, management or ecosystem? Management is the way herders treat their livestock & grassland & can be described. Ecosystem is much broader & more complex & the description is made up of multiple interacting factors, some related to herders, some to livestock, some to plants, some to soil, weather; all interacting & complex. I believe farm size per se has little impact on the ecosystem, except as it affects management. All grassland farms will present a risk to the ecosystem. All need to be managed using methods that emphasize the sustainability of the grassland. Line 88 is, in my mind, irrelevant since all human activity presents a risk to the environment. So, the objective? In a concise form. The objective is a key component of you article & must be precise, clear, concise & avoid ambiguities. I do not find a statement of objectives adequately expressed in lines 83 to101.
Response 8: For the terminology translation of some policies implemented by the government, we have specifically consulted the official literature, revised the "The farm contract responsibility system" to "rural land contract system", and made a brief explanation of the main contents of the policy in the new version. We added special explanations to the “Measures for the Balanced Management of Grass and Livestock” policy, pointing out that: This (policy) has been implemented nationwide since 2005, and the main management objects are farms. In 2020, this policy was suspended temporarily. The provincial governments have formulated local management policies in accordance with the basic principles of previous management policy.
I quite agree with what you said in your email: Ecosystem is much broader & more complex & the description is made up of multiple interacting factors, some related to herders, some to livestock, some to plants, some to soil, weather; all interacting & complex. I believe farm size per se has little impact on the ecosystem, except as it affects management. This sentence makes our research logic clear. What we’re trying to say in our manuscript is that farm size affects the grassland by affecting the management of herders. We have clearly found the impact of farm size on grassland for many years in pastoral research area, but we were unable to define this impact. Through your hints, we can be clear state it. When revising the manuscript again, we have rewrote the paragraph (lines 83 to 101). we tried to make it much shorter and more accurately. We rewrote it several times, which is really difficult than writing longer. The statements on material & methods in this paragraph have been deleted.
We also revised the abstract and some other places of the manuscript accordingly.
In the discussion section, we added a specific discussion on the overgrazing risk of large farm, which is not only related to the current policy, but also provides suggestions for the new policy being adjusted. The added content is ”As the main management objects of the “Measures for the Balanced Management of Grass and Livestock” policy are farms rather than larger areas, overgrazing beyond the farm scope cannot be monitored. Especially since 2020, the provincial governments have formulated local management policies, and the grass transportation industry is developed, the overgrazing across provinces is also a potential risk. Therefore, it is still necessary to carry out unified management nationwide, and the quantitative method of overloading is necessary to investigate a wider space except farm scope. ”
Point 9: I wish to have a well written statement of objective(s). This should be one or two sentences long. It must clearly state the basic or central reason explaining why you carried out your research. Please send this to me with nothing more.
Response 9: The research objective has been changed as:
The overall objective is to analyze the differences and impacts of grazing management related to farm size. The specific objective was to compare economic management behaviors (financing, grassland circulation, and overgrazing management) and utilization of production factors (grasslands , livestock capital, and husbandry labor) between large farms with more than 500 animal unit and small farms less than 500 animal unit.
Point 10: You place grazing management as central to your overall objective. However, in your specific objective you then include economic management behaviors & utilization of production factors as central. In both overall & specific objective you are comparing large & small farms.
Just because I sent you an example of my own does not necessarily mean I think you should have both an overall & a specific objective. Now I do not understand well what you mean by “economic management behaviors & utilization of production factors” but I think “grazing management” is only one factor in “economic management behaviors & utilization of production factors”. If that is the case then it would make more sense to me to put “economic management behaviors & utilization of production factors” as the overall objective & “grazing management” as the specific objective, but I wonder if you need to refer to grazing management except as one factor (see below).
Next, allow me to discuss the phrase “economic management behaviors”. When I search in Google for this phrase I obtain, first, “Behavioral economics” a mixture of economics & human behavior. I also find Behavioral Economics & Management. However, since I am not an expert in behavior, economics or management I suggest you use either economic behavior or economic management, not both.
Now, allow me to discuss “utilization of production factors”. This is clear to me. There are multiple factors that affect production, soil quality, soil management, kind of grassland, animal units & feed source, grassland management, grazing management etc. I have put grazing management as just one factor within your specific objective. This is why I do not understand how grazing management can be in your overall objective. Grazing management is not necessarily the most important factors influencing utilization of production at the level of farm or landscape.
I think economics & farm management are more important in your research than grazing management. In these areas of research I am no help to you. I assume you base your research on the fact that there are differences in grazing management between large & small farms but you do not need to describe what these differences are. If my assumptions are correct then I think you may complete the paper with short mention of grazing management.
Response 12: Thank you for your email of Jan 20, and thank you very much for your detailed communication . I carefully read and thought about your suggestions, please allow me to explain more.
First, my idea of this manuscript is based on the field investigations and herders' interviews in pastoral areas of northern China. Herders regard farm size as a very important factor in production and are sensitive to it. They have many complaints about it. They pointed out that there are obviously different behaviors (economic behaviors) and management ways (grazing management) between large and small farms, and they believed that the differences of farm size had caused unsustainable behaviors and consequences. We wrote this manuscript to present this issue and hope to attract the attentions of researchers and managers. In China's agricultural production areas, there are many studies on the scale of farms (farm size) and its influences, and optimum-scale farm management has been accepted by more people (researchers, managers and producers). However, under the same policy background, the scale of grassland farm and property rights of grassland have not been paid enough attention. Ecologists focus on the grazing rate, and economists rarely get involved in it. In my opinion, grassland farm size should be specially analyzed, because this is an important issue at the level of farm or landscape . You said “economics & farm management are more important in your research than grazing management” , which is determined by our research perspective (C interface).
As you said, good managers may be able to manage grassland sustainably at a higher carrying capacity. I am glad for you brought up the question. Previous studies in China believed that it is easier to implement sustainable management or higher carrying capacity in relatively large farms, but we find that risks were also accompanied. This is one of our research values. After years of observation and investigation, we found that there are not only strong management capabilities but also potential risks in large farms and farm expansion process (encouraged by the government), and we explained this in the conclusion section of this manuscript and put forward our views and suggestions.
We are really troubled when writing research objectives. I inspired by the example you sent us, so changed one objectives to overall objective and specific objective. Now it seems that this may cause misunderstanding, so I changed again, that is, the objective was to compare economic behaviors (financing, grassland circulation, and overgrazing management) and utilization of production factors (grasslands , livestock capital, and husbandry labor) between large farms with more than 500 animal unit and small farms less than 500 animal unit. Grazing management were mentioned in discussion and conclusion sections.
The second explanation is about the application of phrases “economic management behaviors”. “Economic behaviors” is a economic term and we used it before. I added “management” because I want to emphasize that this is the behavior related to management. Now I change “economic management behaviors” to “economic behaviors” according your suggestion.

This manuscript is a resubmission of an earlier submission. The following is a list of the peer review reports and author responses from that submission.
Round 1
Reviewer 1 Report
This manuscript, "Innovative Grassland Resource Management Based on Household Pasture Scale in Inner Mongolia, China" by Xin He et al., presents a very interesting insight into the pasture situation in Mongolia. Many European countries have already experienced this situation which has led in many cases to the abandonment of land. The form of cooperation between small families must be cultivated and encouraged to meet costs and increase profits.
The manuscript deserves to be published, it is well written, very clear in the construction and presentation of the data, and highlights very current issues.
Author Response
Thank you for your good evaluation to our manuscripts.

Reviewer 2 Report
Comments on:
Innovative Grassland Resource Management Based on Household Pasture Scale in Inner Mongolia, China
1. I am concerned that the title has unnecessary or not well understood words. There is no need for the word ‘Resource’. Grassland Management is sufficient. The word ‘Household’ applied to pasture is an unusual descriptor & the meaning is unclear. “Scale’, is a word with too broad a meaning. Does it mean size? Area? Stocking rate? how managed? Pasture is usually understood to be an area of grassland that is grazed.
2. I am also concerned that none of the authors appear to have experience in pasture management or soil science. The consequence of this is that they are not familiar with the English words used internationally & makes this part of the paper difficult to understand.
3. Introduction: line (L) 39 unreasonable is a subjective word. I suggest unsustainable. L 40 Grassland degradation & soil degradation need to be treated separately. L 46 ‘ecological management policies’ ecology cannot be managed, the ecosystem can be. Ll 52 to 94 are impossible for me to understand. Words are used incorrectly (for example grassland-animal interface & grassland livestock management, household production behavior, ecological risk of grassland, grassland resource utilization behavior), words are used which do not have a specific meaning (for example scale, different livestock scale levels, industrialization of animal husbandry, large and small household pastures, the ecological landscape of grasslands has become segmented, modern animal husbandry, grazing intensification and large-scale livestock production, quantity control of grassland resource management policy)
4. Ll 78 to 94 are a summary of past research. A concise understandable statement of objectives of the research to be reported in this article is required.
Author Response
Thank you for your suggestions, please see the attachment.

Reviewer 3 Report
Manuscript ID: agriculture-1828175Review for Agriculture
Title: Innovative Grassland Resource Management Based on Household Pasture Scale in Inner Mongolia, China. Authors: Xin He, Jingru Wei, Suhua Gu, Luping Wang, Zechen Tian and Danqiong ChenGeneral comments:
1. This manuscript has important and interesting information on the impact of household pasture scale on grasslands from the perspective of livestock. Behaviors between large and small household pastures in terms of financing, land circulation, and overstocking management are discussed.
2. The aim of this study was to analyze the use of grassland resources in the household pasture production.
3. The data was collected from 2004 to 2020, using a total of 126 household pastures in Inner Mongolia, while carrying out an analysis of the limitations of livestock scale level.
4. The results show a trend towards implementation of larger and more modern pastures, which is more effective economically.
5. In my opinion, a paper based only on the financial behavior issue of livestock householders does not fit this journal.
6. The manuscript is not written well and needs much work by an editor to improve the English.
7. Conclusions and the list of references are missing.
8. The manuscript should be rejected.
9. Because of its interesting data I would suggest the authors, after rewriting the paper, to submit it to a journal that deals more with economic and social issues.
Other comments:
Title: The financial matter should be mentioned in the title
Abstract: Written well and clear, but is misleading because of the ecological point of view that is expected from this paper and not examined and discussed.
Keywords:
OK.
Introduction
*This chapter needs much work by an editor to improve English.
*Lines 73-76: “In the field investigation…… of household pasture”. - This should be mention in the “Results” chapter not in the “Introduction”.
Materials and Methods
*Also this chapter needs much work by an editor to improve English.
*The use of the term “statistical data” which is mention in many places is not correct. “statistical analysis” could be used.
*Delete lines 134-136, “…and the continuous data points of 126 household pastures in the grassland could be obtained in this study”. It was mentioned before.
Results
*This chapter aims to present the result collected in this study. There are parts that belong to the “Discussion”, including chapter 3.3. “Differential Mechanisms of Grassland Resource Use”.
*References should not be mentioned in the “Results” chapter.
Discussion
*As the benefit for larger farms was shown in this study, it has to be connected, related, mentioned (cited) by other studies that were conducted in China and over the world.
*Authors must enlarge this chapter
Conclusions
*Add “Conclusions” chapter.
*No references were attached.
Author Response

(The authors gave the same response as above.)

Reviewer 4 Report
A very interesting research work,This study combined field investigations and statistical analysis from 2004 to 2020, using a total of 126 household pastures in Xilinguole League of Inner Mongolia in China,The scope of the survey and research is large, the data obtained is valuable, can be good to know the actual production, and can provide policy support to the local government, statistical analysis is also relatively good, there are the following recommendations need to be revised.
1. The language need further edited by the English native speaker. Many sentences are incomprehensible and can only be guessed.
2. The abstract section, tells a lot, but I would like to see the most important results of your research and whether you can give possible advice to small household pastures.
3. In the introduction section, there is a lot of research background on Inner Mongolia Grassland, has anyone done similar research studies, and what is the progress of your research compared to the previous studies.
4. Add the conclusion section to highlight your most important research rebates, the many disadvantages of farm owners facing financial risk and grass degradation, and whether you have responsive recommendations.
Author Response

(The authors gave the same response as above.)
